# TGFβ1 Induces Senescence and Attenuated VEGF Production in Retinal Pericytes

**DOI:** 10.3390/biomedicines10061404

**Published:** 2022-06-14

**Authors:** Dragana Avramovic, Sébastien A. Archaimbault, Alicia M. Kemble, Sabine Gruener, Mirjana Lazendic, Peter D. Westenskow

**Affiliations:** 1Ocular Technologies, Immunology, Infectious Diseases and Ophthalmology, Pharmaceutical Research and Early Development, Roche Innovation Center Basel, F. Hoffmann-La Roche Ltd., 4070 Basel, Switzerland; sebastien.archaimbault@roche.com (S.A.A.); sabine.gruener@roche.com (S.G.); mirjana.lazendic@roche.com (M.L.); 2Neuroscience and Rare Disease, Pharmaceutical Research and Early Development, Roche Innovation Center Basel, F. Hoffmann-La Roche Ltd., 4070 Basel, Switzerland; alicia.kemble@gmail.com

**Keywords:** diabetic retinopathy, vascular retinopathy, microvascular disease, pericyte dropout, VEGF, TGFβ1, IL6, senescence, angiogenesis, endothelial junctions

## Abstract

Diabetic retinopathy (DR) is a microvascular disease of the retina and a serious complication of type I and type II diabetes mellitus. DR affects working-age populations and can cause permanent vision loss if left untreated. The standard of care for proliferative DR is inhibiting VEGF. However, the mechanisms that induce excessive VEGF production in the retina remain elusive, although some evidence links elevated VEGF in the diabetic retina with local and systemic TGFβ1 upexpression. Here, we present evidence from animal models of disease suggesting that excessive TGFβ1 production in the early DR is correlated with *VEGF* mRNA and protein production by senescent pericytes and other retinal cells. Collectively, these results confirm that TGFβ1 is strongly implicated in the vascular complications of DR.

## 1. Introduction

Diabetic retinopathy is a complication of diabetes mellitus characterized by alterations in the retinal vasculature [1]. The retina is one of the most vascularized and highest energy-demanding tissues of the human body [2], and vision and retinal homeostasis depend on blood–retinal barrier integrity [3,4]. Alterations on diabetic retinal neurons induce microvascular complications that cause tight junction instability, basement membrane thickening, pericyte dropout, and formation of acellular capillaries. These effects collectively induce ischemia, excessive vascular permeability, microaneurysms, and neovascularization [5,6].

There are two blood–retinal barriers (BRBs) in the retina. The inner blood–retinal barrier (iBRB) supports photoreceptors and interneurons, and is the vasculature most impacted by diabetic retinopathy [7,8]. Tight endothelial junctions and adjacent pericytes ensure barrier stability. Pericyte loss is recognized as one of the earliest structural changes [9,10], as the endothelial cell to pericyte ratio drops from 1:1 in healthy retinas to 4:1 in diabetic retinas. While pericyte dropout is considered to be a critical inducer of vascular instability, the mechanisms inducing dropout are poorly understood [11,12,13]. Likewise, the mechanism linking high glucose levels and pericyte instability remains unclear.

Loss of vascular capillaries leads to retinal hypoxia and release of vascular endothelial growth factor (VEGF). VEGF has a central role in the pathophysiology of diabetic retinopathy and is identified as both a predictive marker of disease progression and the primary driver of diabetes-induced vasculopathy [14,15]. IL6 is an additional factor that is elevated in ocular fluids of DR patients, and its levels correlate with disease severity. Furthermore, IL6 is recognized as a biomarker of retinal inflammation [16], and reportedly is a critical player in oxidative-stress-induced inflammation and dysfunction of diabetic neurovascular units [17,18]. Finally, studies have shown that IL6 induces BRB instability by downregulating tight junction proteins and upregulating inflammatory cytokines [19,20].

Another key effector factor in diabetic retinopathy may be TGFβ1. TGFβ1 is elevated in total serum of diabetic patients [21] and its levels correlate with diabetic retinopathy disease progression [22]. Furthermore, elevated TGFβ1 levels correlate with increased levels of VEGF [22]. Studies have shown that TGFβ1 levels are sensitive to high glucose levels in humans [23], in animals [24], and in vitro [25] and that different pharmacological interventions used in the treatment of diabetes reduce TGFβ1 total serum levels [26,27]. Additionally, the role of TGFβ1 in diabetic nephropathy is well described where TGFβ1 plays a role as a prognostic marker and an active player in disease progression [24,26,27,28].

TGFβ1 is a pleiotropic and highly conserved growth factor that plays a role in various cellular functions during development and homeostasis [29,30,31]. Dysregulation of TGFβ expression is an essential driver of multiple diseases, such as fibrosis, cancer, immune disease, and vasculopathies [30,32,33,34,35], where it exerts differential functions in a context- and tissue-dependent manner. There are three isoforms of the TGFβ molecule: TGFβ1, TGFβ2, and TGFβ3. TGFβ1 is the most abundant and predominantly expressed in endothelial, hematopoietic, and connective tissue cells [32,35]. TGFβ signals via TGFβ type I and II receptors that are surface-bound serine/threonine protein kinases. TGFβ binds TGFβ type II receptor (TβRI), leading to heterodimerization with TGFβ type II receptor (TβRII) and phosphorylation of threonine and serine residues in the TTSGSGSG motif of TβRI and its activation [30,31,32]. Activated TβRI recruits and phosphorylates the Smad2/3 proteins, forming a heterocomplex with Smad4 that translocates to nuclei and regulates the transcription of the target genes [36,37].

Furthermore, TGFβ1 is implicated in the process of cellular aging, where it induces senescence in various cells and tissues [38] via expression of tumor suppressor genes, among others, p21Cip1 (p21) and p15Ink4b (p15) [39].

Senescence was first reported in 1961 by Hayflick et al. [40], and was defined as degeneration after about 50 sub-cultivations and one year in culture. It has initially been attributed to the dysregulation of unknown intrinsic factors at the cellular level. Aging is characterized by accumulating senescent cells in multiple tissues over time. At the cellular level, senescence is a stress-induced and multistep process characterized by G1 growth arrest, senescence-associated morphological changes, senescence-associated heterochromatin foci (SAHFs), SA-β-Gal activity, and a senescence-associated secretory phenotype (SASP) [41,42,43].

Besides alterations in TGFβ1 and VEGF, senescence in retinal cells may contribute to diabetic retinopathy pathogenesis. Furthermore, elevated TGFβ1 levels may be responsible, at least in part, for both microvascular and senescent phenotypes in diabetic retinopathy. Moreover, TGFβ is elevated in DR patients. Together, elevated VEGF expression and pericyte dropout are key characteristics of DR. Based on these observations, we set out to determine if/how TGFβ could affect pericyte homeostasis and VEGF levels in vitro and in an animal model of disease.

In this study, we report that exogenous TGFβ1 can induce VEGF expression in primary retinal pericytes and in mouse retinas. In addition, TGFβ1 can induce senescence in retinal pericytes in vitro and increased expression of VEGF, IL6, and an SASP potentially via p15 and PAI1 pathways. Collectively, these findings suggest that TGFβ1, which is compensatory upregulated in aging populations, could be an important causative and disease-propagating factor in diabetic retinopathy pathogenesis.

## 2. Materials and Methods

### 2.1. Cell Culture

Primary human retinal pericyte cells (ACBRI 183, Cell Systems, Seattle, WA, USA) were maintained in complete medium (4Z0-500, Cell Systems) and cultured at 37 °C in a humidified incubator with an atmosphere of 95% O2 and 5% CO_2_.

### 2.2. Reagents

Recombinant human TGFβ1 protein (240-B-002/CF), human TGFβ2 protein (302-B2-002/CF), human TGFβ3 protein (243-B3-002/CF), recombinant mouse TGFβ1 protein (7666-MB-005/CF), and mouse TGFβ2 protein (7346-B2-005/CF) were obtained from R&D and reconstituted as recommended by the manufacturer. Recombinant mouse TGFβ3 protein (SRP6552-5UG) was purchased from Sigma (St. Louis, MO, USA).

### 2.3. Senescence Assay

An assay kit from Abcam (ab228562) was used to identify senescent cells. Pericytes were seeded (5 × 10^5^ cells per well) in a 24-well plate and incubated for 24 h, and then treated with TGFβ1 and incubated for 24, 48, and 72 h. After the treatment, the media was removed and replaced with fresh media containing 1.5 µL of senescence dye (ab228562, Abcam, Cambridge, UK) in 500 µL of fresh media. After incubation for 1 h at 37 °C, 5% CO_2_, the cells were washed two times with 500 µL wash buffer (ab228562, Abcam). Cells were collected by trypsinization and resuspended in 500 µL wash buffer and analyzed for Alexa488 signal immediately using flow cytometry (BD Fortessa LSR, Ashburn, VA, USA).

### 2.4. MTT Assay

MTT cell proliferation assay kit from Abcam (ab211091) was used to quantify cell proliferation. Cells were plated in a 96-well plate and treated with TGFβ1 for 24, 48, and 72 h. After the treatment, the media was replaced with 50 µL of serum-free media and 50 µL of MTT reagent (ab211091, Abcam) per well. After incubation at 37 °C for 3 h, MTT solvent (ab211091, Abcam) was added to each well. The plate was protected from light and shaken on an orbital shaker for 15 min. The absorbance at 590 nm was recorded using a plate reader (Infinite^®^ 200 PRO, Kawasaki, Japan).

### 2.5. LDH Assay

The cytotoxicity detection kit from Sigma Aldrich (St. Louis, MO, USA) (11644793001) was utilized according to the manufacturer’s instructions. Briefly, 100 µL of supernatant from treated cells and 100 µL of freshly prepared reaction mixture were mixed in a 96-well plate and incubated for up to 30 min at room temperature protected from light. The absorbance was measured at 490 nm.

### 2.6. Luminex Assay

The supernatants from TGFβ1-treated pericytes were analyzed with Luminex multiplex technology. We custom selected cytokines from R&D and utilized Luminex200 to detect levels of TNFα, IL6, IL8, GM-CSF, EGF, VEGF, bFGF, PLGF, ANGPTL4, ICAM1, Collagen I, Collagen IV, Fibronectin, MMP1, MMP3, and TIMP1, and the results were analyzed with the software Bio-Plex Manager (Version 6.2.0175). Luminex technology was also used to assess protein levels of cytokines in ocular fluid collected from TGFβ1-, TGFβ2-, and TGFβ3-injected mice. For these studies, a customized Luminex multiplex immunoassay panel for mice cytokines was generated and purchased from R&D.

### 2.7. QPCR

Total RNA was isolated with RNeasy Mini Kit (74106, Qiagen) and measured with nanodrop (NanoDrop™ One, Thermo Scientific, Waltham, MA, USA). IScript™ cDNA Synthesis Kit (1708891, BioRad) was used for cDNA synthesis. Primers for VEGF (Hs00900055_m1), IL6 (Hs00174131_m1), PAI1 (Hs00167155_m1), and p15Ink4b (Hs00793225_m1) were obtained from TaqMan (4331182, Thermo Fisher Scientific), and β-actin primers were obtained from Sigma (05046165001). Quantitative PCR was performed using TaqMan Fast Advanced Master Mix (4444557, Applied Biosystems™, Waltham, MA, USA) and The Applied Biosystems QuantStudio 12K Flex Real-Time PCR System. Further, the following protocol was utilized: denature for 2 min at 95 °C, followed by 40 cycles, each consisting of 95 °C denaturation for 1 s, and 60 °C annealing for 20 s. The relative expression level of the transcripts was analyzed using the 2-△△Ct approach.

### 2.8. ELISA

Supernatant from TGFβ1-treated cells were used to detect cytokines of interest. The kits from Invitrogen (BMS277-2, BMS213-2, and BMS2033) for the detection of human VEGF, IL6, and PAI1, respectively, were used. A Tecan (Infinite^®^ M1000 PRO) was utilized for absorbance detection.

### 2.9. TGFβ-Injected Mice

C57BL/6 mice were intravitreally injected with mouse TGFβ1, TGFβ2, and TGFβ3. After 48 h, mice were euthanized, corneas were removed, and the eyes were collected. One eye of each animal was used for ocular fluid collection that was further analyzed with Luminex. The other eye was collected and paraffin-embedded following the protocol described below.

### 2.10. Fixation and Paraffin Embedding for Histology

The eyes were fixed in 10% NFB (J.T.Baker, 3933.9020) overnight and washed in PBS. We used the dehydration machine (Tissue-Tek VIP^®^ 5 Jr, Sakura Finetek, Alphen aan den Rijn, The Netherlands), program one, for dehydration of mice eyes. In addition, eyes were embedded in paraffin.

### 2.11. ISH Staining

Glass slides (SuperFrost Plus, VWR, 631-0108, Waltham, MA, USA) were dried and washed in xylene (Sigma, 95672) in advance. Eyes were cut into 5 µm sections with a Leica microtome (RM2255) and mounted onto the glass slides in mounting media (EUKITT^®^, 01-0600). Ventana system, ACD probes, and ACD kits were used for automated ISH. ACD probe used: RNAscope^®^ 2.5 VS Probe- Mm-Vegfa-O1 Cat No. 436969.

Images were acquired with slide scanner VS120 Virtual Slide Microscope (Olympus, Tokyo, Japan) and the signal was quantified with HALO software (V3.2).

### 2.12. Statistical Analysis

Each experiment was repeated at least three times unless stated otherwise. Prism 7 (GraphPad) was used to perform statistical analyses and generate charts. Statistical analysis included unpaired, Student’s *t*-test, one-way ANOVA, or two-way ANOVA. Furthermore, applicable post hoc correction tests (Bonferroni procedure, Tukey’s Test, Šídák method, and Dunnett’s correction) were executed. For all graphs, data are presented as mean ± SD. *p* values < 0.05 were considered significant.

## 3. Results

### 3.1. TGFβ1 Induces Senescent Features in Primary Retinal Pericytes

Based on the hypothesis that *TGFβ1* could induce defects in retinal pericytes, we performed FACS-based senescence assays. We cultured primary retinal pericytes and treated them with 50 ng/mL of TGFβ1 for 24, 48, and 72 h. After these timepoints, we set out to quantify production of senescence-associated β-galactosidase (SA-β-gal), which is detectable only in senescent cells (Figure 1a,b). Using this technique, we detected elevated numbers of SA-β-gal+ cells upon treatment, suggesting that TGFβ1 induces senescence in retinal pericytes. Moreover, to exclude any cytotoxic contribution of TGFβ1 on pericytes, we also used MTT and LDH assays to measure proliferation and cytotoxicity. Collectively, these data suggest that TGFβ1 treatment induces SA-β-gal production while slowing cell proliferation and, importantly, not inducing cellular toxicity (Figure 1c).

Next, pericytes were treated with 50 ng/mL of TGFβ1 for 24 h, and supernatants were collected for Luminex analysis. The results from the Luminex assay showed an increase in SASP elements in the supernatant of TGFβ1-treated pericytes. Primarily, we detected high levels of VEGF, IL6, and ANGPTL4 (Figure 1d).

### 3.2. TGFβ1 Regulates na SASP, p15, and PAI1 in Senescent Retinal Pericytes 

To understand the dynamics of how pericytes respond to TGFβ1 treatments, we performed a time-course experiment. Therefore, we treated retinal pericytes as described and collected samples for qPCR and ELISA at different time points. qPCR reflected changes at the gene level, whereas ELISA revealed modulation on the protein level. An increase in gene expression was transient, reaching a maximum after six hours. On the other hand, changes in protein expression required 24 h, as anticipated (Figure 2a,b). 

Cellular senescence is initiated by cell cycle arrest driven by the expression of cell cycle regulators, such as p53, p21, p16, and p15. In addition, plasminogen activator inhibitor-1 (PAI-1), a pleiotropic factor under the regulation of TGFβ, reportedly regulates cell cycling. Therefore, we analyzed the expression patterns of cell cycling factors and found that p15 and PAI1 are upregulated upon TGFβ1 stimulation (Figure 2c), whereas the expression of neither p21 nor p16 was changed (data not shown).

Collectively, our data suggest that TGFβ can control multiple molecular pathways of senescence in pericytes. Furthermore, we speculate that this could be an early step in diabetic retinopathy disease progression that contributes to pericyte dropout.

### 3.3. TGFβ1 Induces VEGF Expression in Mice Retina

Since local and systemic TGFβ1 increases are detectable in diabetic patients, understanding the local effect of TGFβ1 in the retina is essential. To determine the impact of TGFβ1 protein in vivo, animals were intravitreally injected with 50 ng/mL of TGFβ1, and eyes were collected after 48 h for ocular fluid analysis with Luminex, and tissue was paraffin-embedded for ISH analysis. VEGF protein levels detected via Luminex were increased in TGFβ1-injected animals after 48 h (Figure 3a). In addition, TGFβ-injected eyes sectioned for gene expression analysis demonstrated a co-ordinated increase at mRNA levels of VEGF upon TGFβ1 treatment, whereas TGFβ2 and TGFβ3 treatment did not show the same effect (Figure 3b). Interestingly, VEGF expression appeared highest in the inner nuclear layer (INL). 

Local expression of VEGF within retinal neurovascular units could be a major contributing driver of microvascular complications in disease progression. 

## 4. Discussion

The prevalence of diabetes has been steadily increasing over the past few decades [44,45]. Diabetic retinopathy is a life-altering complication of diabetes that affects working-age populations [1,6,9].

The mechanism by which high blood glucose levels induce progressive damage of the retinal vessels remains unclear. Here, we propose an early role of TGFβ in the cascade of structural and functional alterations in the endothelium that eventually leads to retinal dysfunction. Glucose increases activation of TGFβ [33,46] and, consequently, TGFβ levels increase systemically and locally in diabetic animal models [47] and in patients [21,22,27,33]. TGFβ is also recognized as a biomarker and significant player in some complications of diabetes [24,28,48,49]. Likewise, TGFβ1 levels are elevated systemically and in aqueous humor in patients with diabetes [50] and diabetic retinopathy [51,52,53]. Thus, understanding the local cellular response to elevated TGFβ is highly relevant. 

In the retina, TGFβ signaling plays an essential part in retinal vasculature development, verified in the conditional knock-out TβRIIΔeye mice. The authors detected impaired retinal capillaries exhibiting torturous, leaky vessels, microaneurysms, and hemorrhages [54]. Beyond development, the TGFβ axis plays a role in the stability of matured vessels. Based on published evidence, TGFβ2 is a predominant isoform in the eye and, as such, it plays a role in vessel stabilization and prevents angiogenesis in homeostasis. Interestingly, an active form of TGFβ1 is absent in the aqueous and vitreous humor of healthy eyes and the ratio among the three active isoforms is 1:0.4:0 for β2:β3:β1 in the aqueous humor of healthy human eyes [55]. TGFβ2 is also predominant in retinal pigment epithelium (RPE) [24,56,57].

TGFβ is implicated in pathological angiogenesis in the retinal vasculopathies; however, the mechanism is not fully understood. Contrasting published data support both antiangiogenic and proangiogenic roles of TGFβ1 [58,59]. 

Here, we have shown that TGFβ1 stimulates VEGF expression in vitro and in vivo. Further, we hypothesize that a mechanism by which TGFβ1 induces the changes in the diabetic retina could be cellular senescence. In cultured retinal pericytes, TGFβ1 induces characteristics of cellular senescence, including cell cycle arrest, SASP production, and morphological changes. Moreover, we show that this effect possibly occurs via the regulation of p15 and PAI1 genes. This is in agreement with the reported role of TGFβ in cell cycle arrest where TGFβ signaling stimulates p21, p15, and p57 expression in cancer cells [60].

We have also demonstrated that TGFβ1 induces senescence in retinal pericytes for the first time. Therefore, we propose that cellular senescence plays a role in early events of diabetic retinopathy, such as pericyte dropout, angiogenesis, and immune cell infiltration. Furthermore, we show that TGFβ1 induces, besides VEGF, the release of many essential cytokines that play a role in the progression of diabetic retinopathy. For example, IL-6 is recognized as an inducer of retinal vascular inflammation and permeability [16]. ANGPTL-4 induces diabetic retinal inflammation [61], whereas PLGF is recognized as a homolog of VEGF and a player in diabetic retinopathy [51,62]. Finally, in TGFβ1-injected eyes, we have detected increased levels of VEGF at both gene and protein levels, confirming the effects of TGFβ1 we reported in vitro. 

## 5. Conclusions

Our data suggest that TGFβ1 induces VEGF production and release in vitro and in vivo. In retinal pericytes, this effect is via the mechanism of cellular senescence, and it further elevates other cytokines that play a role in the development of diabetic retinopathy. Therefore, we have proposed that TGFβ1-induced senescence may play a role in the disease progression.

We have conducted this pilot study that will be extended in the future; nevertheless, our results strongly suggest that TGFβ should be further studied as an early player in diabetic retinopathy. Furthermore, identifying other early players in the disease should be elucidated in future studies, as early underlying processes remain elusive. Finally, the contribution of senescence in DR phenotypes has become more evident with recent studies; therefore, we believe that studying the involvement of this process in diabetic retinopathy might shed new light on overlooked drivers and mechanisms of this devastating disease. 

## Figures and Tables

**Figure 1 biomedicines-10-01404-f001:**
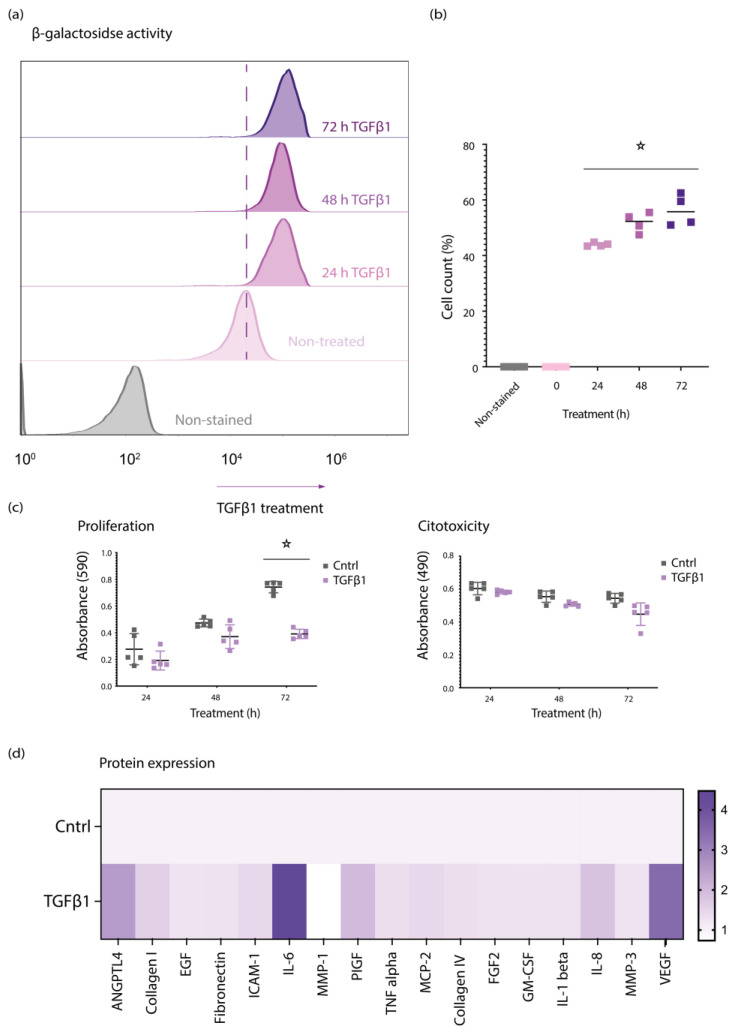
TGFβ1 induces cellular senescence in primary retinal pericytes. (**a**) In senescence assay, retinal pericytes were treated with 50 ng/mL of TGFβ1 for 24, 48, and 72 h and β-galactosidase activity was assessed by flow cytometry. The shift towards right indicates increased Β-galactosidase activity; (**b**) quantification of flow cytometry data; (**c**) MTT assay shows decreased proliferation and LDH shows no effect on viability of cells treated with 50 ng/mL of TGFβ1; (**d**) Luminex assay shows increase in SASP elements upon 50 ng/mL of TGFβ1 for 24 h. Graphs show mean and standard deviation (±SD). All experiments were performed at least in triplicate, and asterisks represent *p* < 0.05.

**Figure 2 biomedicines-10-01404-f002:**
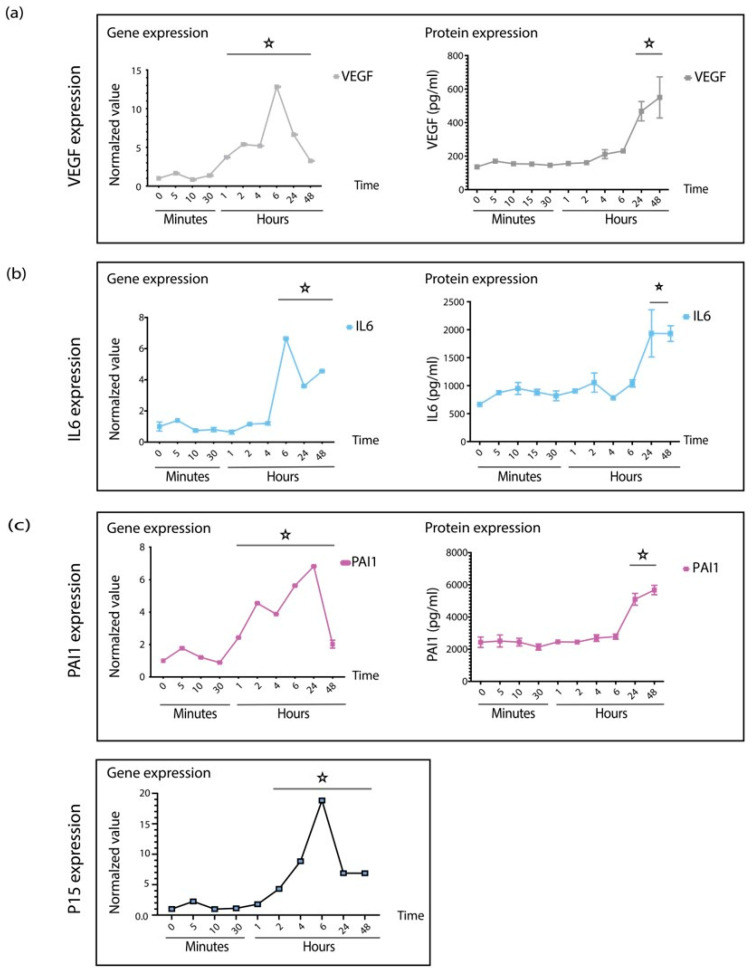
TGFβ1 regulates senescence at the gene level. (**a**) QPCR and ELISA data demonstrate the expression patterns of VEGF and (**b**) IL6 at gene and protein levels. (**c**) TGFβ1 treatment elevates expression of PAI and P16. Graphs show mean ±SD. All experiments were performed at least in triplicates, and asterisks represent *p* < 0.05.

**Figure 3 biomedicines-10-01404-f003:**
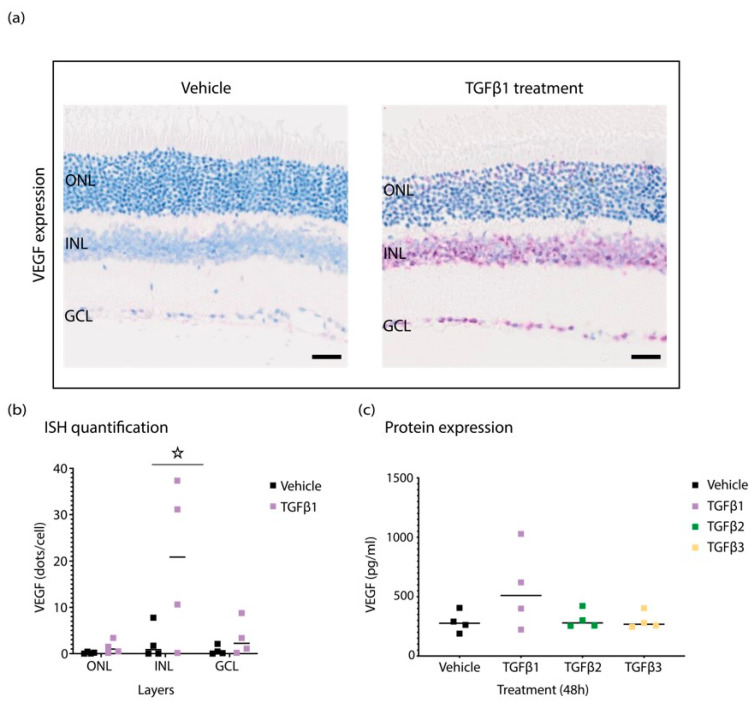
Retinas from TGFβ1-injected eyes show increased levels of VEGF expression. (**a**) Representative ISH staining shows elevated levels of VEGF in TGFβ1-treated retina in comparison to vehicle-injected controls. (**b**) Quantification of ISH staining with HALO. (**c**) The analysis of cytokine levels with Luminex shows increase in VEGF expression at the protein level in ocular fluid of TGFβ1-injected eyes. Graphs show mean ± SD. All experiments were performed with four replicates, and asterisks represent *p* < 0.05.

## Data Availability

Not applicable.

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
