# Peer review of "TGFβ1 Induces Senescence and Attenuated VEGF Production in Retinal Pericytes"

_biomedicines, 2022, doi:10.3390/biomedicines10061404_

Round 1

Reviewer 1 Report

Avramovic et al. realized a very interesting article describing how “TGFb1 induces senescence and attenuated VEGF production in retinal pericytes”. I consider the manuscript very fascinating but, at the same time, I suggest several revisions needed to improve the reliability and the completeness of the paper:

·    Globally, I suggest the authors to underline that their obtained results are “pilot” ones, and they should be extended in the near future.

·         The “Introduction” section did not speak sufficiently about retinal compromission related to inflammation and vascular alterations, targets of authors’ work. Thus, I suggest the authors to add more recent references related to inflammation, oxidative stress, and angiogenesis in association to retinal degenerations. The recent PMID: 34440511 and PMID: 34058230 could represent a substrate able to enforce the role of considered cellular mechanisms.

·         Are experiments realized at least in triplicate?

·         In “QPCR” section the list of used primers should be added.

·         The “Statistical Analysis” section lacks post-hoc correction test (e.g. Bonferroni).

·         Finally, manuscript requires English revisions and typos correction.

Reviewer 2 Report

REVIEWER’S COMMENTS

The manuscript TGFb1 induces senescence and attenuated VEGF production in retinal pericytes.by Avramovic et al confirmed that TGFβ1 is involved in the vascular complications of DR.

1.     Please include the catalog numbers and manufacturer(s) of all the reagents used in this study.

2.     Please include “n” and explain the asterisks/p-values in the figure legends.

3.     Please improve the resolution of the figures. Resolution should be at least 300 dpi.

4.     Please briefly discuss the potential future directions that could be a follow-up to this study.

5.     Please be consistent with the style of references.

Round 2

Reviewer 1 Report

The manuscript can to be accepted in the present form